# HC-GAE: The Hierarchical Cluster-based Graph Auto-Encoder for Graph Representation Learning

**Lu Bai**[1,2], **Zhuo Xu**[1], **Lixin Cui**[3]*, **Ming Li**[4,5], **Yue Wang**[3], **Edwin R. Hancock**[6]

[1]School of Artificial Intelligence, Beijing Normal University, Beijing 100875, China
[2]Engineering Research Center of Intelligent Technology and Educational Application,
Ministry of Education, Beijing Normal University, Beijing 100875, China
[3]School of Information, Central University of Finance and Economics, Beijing 100081, China
[4]Zhejiang Institute of Optoelectronicss, Jinhua 321004, China
[5]Zhejiang Key Laboratory of Intelligent Education Technology and Application,
Zhejiang Normal University, Jinhua 321004, China
[6]Department of Computer Science, University of York, York YO10 5GH, United Kingdom

## Abstract

Graph Auto-Encoders (GAEs) are powerful tools for graph representation learning. In this paper, we develop a novel Hierarchical Cluster-based GAE (HC-GAE), that can learn effective structural characteristics for graph data analysis. To this end, during the encoding process, we commence by utilizing the hard node assignment to decompose a sample graph into a family of separated subgraphs. We compress each subgraph into a coarsened node, transforming the original graph into a coarsened graph. On the other hand, during the decoding process, we adopt the soft node assignment to reconstruct the original graph structure by expanding the coarsened nodes. By hierarchically performing the above compressing procedure during the decoding process as well as the expanding procedure during the decoding process, the proposed HC-GAE can effectively extract bidirectionally hierarchical structural features of the original sample graph. Furthermore, we re-design the loss function that can integrate the information from either the encoder or the decoder. Since the associated graph convolution operation of the proposed HC-GAE is restricted in each individual separated subgraph and cannot propagate the node information between different subgraphs, the proposed HC-GAE can significantly reduce the over-smoothing problem arising in the classical convolution-based GAEs. The proposed HC-GAE can generate effective representations for either node classification or graph classification, and the experiments demonstrate the effectiveness on real-world datasets.

## 1 Introduction

In real-world applications, graph structure data has been widely used for characterizing pairwise relationships among the components of complex systems. With the recent rapid development of deep learning, the graph representation learning approaches relying on neural networks are introduced for the analysis of various graph data, e.g., social networks [12], transportation networks [20], protein compounds [34], 3D shape [2, 5], etc. One challenging arising in these studies is that the graph data has a nonlinear structure defined in an irregular non-Euclidean space, and it is hard to directly employ traditional neural networks to learn graph representations.

To overcome the above problem, there have been increasing interests to further generalize traditional neural networks, especially the Convolutional Neural Network (CNN) [18], for the irregular graph

---

*Corresponding Author: cuilixin@cufe.edu.cn

38th Conference on Neural Information Processing Systems (NeurIPS 2024).

data. These are the so-called convolution-based Graph Neural Networks (GNNs) [4] and their related approaches [13] proposed for graph-based tasks, utilizing both graph features and topologies. For instance, the Higher-order Graph Convolutional Network (HiGCN) [11] has been developed based on the higher-order interactions to recognize intrinsic features across varying topological scales. Its effective expressiveness makes it capable for various graph-based tasks. The DeepRank-GNN [26] has been proposed by combining the rotation-invariant graphs and the GNN to represent protein-protein complexes. Because the GNN models can extract graph representations with more semantic learning under supervised conditions, researchers have focused more on seeking a self-supervised framework associated with the GNNs to accomplish representation learning.

As a typical framework of representation learning, the classical Auto-Encoder [22] has been proposed to extract impressive results by reconstructing the input information. Especially, the Graph Auto-Encoder (GAE) [30] associated with the GNN model has further generalized the reconstruction ability for graph structures [15, 19]. Due to the extensibility, the GAEs have been developed as a family of classical models for self-supervised representation learning, and there are adequate derivation models belonging to GAEs. For instance, the Self-Supervised Masked Graph Autoencoders (GraphMAE) [9] focusing on the feature reconstruction adopts a masking strategy and the scaled cosine error in the training model. Compared to the traditional GAE approach like VGAE [29], its decoder is retrofitted with the GNN and the re-masking operation. Based on the GraphMAE, the S2GAE [28] continues to adopt the masking strategy to improve the auto-encoder framework. To generate the cross-representation, the decoder is designed to capture the cross-correlation of nodes.

**Challenges.** Although the classical GAE-based methods achieve the effective performance for graph representation learning, they still have some significant challenging problems summarized as follows.

**(a) The limitation for multiple downstream tasks**: Generally, the representations extracted from the GAEs can be divided into several categories, including the node-level representations for node classification, the graph-level representations for graph classification, etc. Specifically, it is difficult for the GAEs to generate universal representations for multiple downstream tasks simultaneously. This is because the GAEs tend to over-emphasize the node features. For instance, the GraphMAE [9] focuses more on the node feature reconstruction, resulting in **topological missing** and weakening the the structure information reconstruction. This is harmful for the graph-level representation learning.

**(b) The over-smoothing problem**: The GAEs are usually proposed based on the GNNS, thus both the decoder and encoder modules of the GAEs are defined associated with a number of stacked graph convolution operations, that rely on the node information propagation between adjacent nodes. When the GAE becomes deeper, the node features tend to be similar or indistinguishable after multiple rounds of information passing [21], resulting in the notorious over-smoothing problem [6] and influence the performance of the GAEs.

**Contributions.** The aim of this paper is to overcome the above challenging problems by proposing a novel HC-GAE model. Overall, the main contributions are threefold.

**First**, we propose a novel Hierarchical Cluster-based GAE (**HC-GAE**) for graph representation learning. Specifically, for the encoding process, we adopt the hard node assignment to decompose a sample graph into a family of separated subgraphs. We perform the graph convolution operation for each subgraph to further extract node features and compress the nodes belonging to each subgraph into a coarsened node, transforming the original graph into a coarsened graph. Since the separated subgraphs are isolated from each other, the convolution operation cannot propagate the node information between different subgraphs. The proposed HC-GAE can in turn reduce the over-smoothing problem arising in the classical GAEs. Moreover, since the effect of the graph structure perturbation is limited within each subgraph, the required convolution operation performed on each subgraph can strengthen the robustness of the encoder for the proposed HC-GAE. As a result, the outputs of the encoder can be employed as the graph-level representations. On the other hand, for the decoding process, we adopt the soft node assignment to reconstruct the original graph structure by expanding each coarsened node into all retrieved nodes probabilistically. Thus, the outputs of the decoder can be employed as the node-level representations. Since the HC-GAE is defined by hierarchically performing the above compressing procedure during the decoding process as well as the expanding procedure during the decoding process, the proposed HC-GAE can effectively extract bidirectionally hierarchical structural features of the original sample graph, resulting in effective hierarchical graph-level and node-level representations for either graph classification or node classification.

**Second**, we propose a new loss function for training the proposed HC-GAE model. For calculating the complete loss value, we integrate the local loss from the subgraphs in the encoding operation and the global loss from the reconstructed graphs in the decoding operation. The global loss can capture the information from both the structure and the feature reconstruction processes. The combination of these two pretext tasks broadens the strict requirement causing the topological closeness. In addition, to avoid the over-fitting problem, we add the local loss as the regularization in our loss function.

**Third**, we empirically evaluate the performance of the proposed HC-GAE model on both node and graph classification tasks, demonstrating the effectiveness of the proposed model.

This paper organizes as follows. Section 2 reviews some related works. Section 3 gives the definition of the proposed model. Section 4 provides experimental evaluations. Section 5 gives conclusions.

## 2 Related Works

### 2.1 The Graph Neural Network

GNNs are widely employed for many real-world applications [16, 31, 35], and have achieved a prominent success. The input data of GNNs is a graph, that is a kind of non-Euclidean data and contains nodes and edges. With the complex structures of the graphs, GNNs aim to leverage the information passing mechanism among nodes for graph embedding learning. The procedure of the information passing can be divided into aggregating, combining and readout. Specifically, given an input sample graph $G(V, E)$ with the node set $V$ ($|V| = n$) and the edge set $E$, the node information is represented as the feature matrix $X \in \mathbb{R}^{n \times d}$ with $d$ features, and the structure information is represented as the adjacent matrix $A \in \{0, 1\}^{n \times n}$. The GNN for each layer is defined as

$$Z_G = \text{GNN}(X, A; \Theta), \tag{1}$$

where $\Theta$ is the parameter set, and $Z_G$ is the graph embedding result for downstream tasks.

The Graph Convolutional Network (GCN) [13] is a typical derivative model of GNNs, and generalizes the classical Convolutional Neural Network (CNN) [18] from the regular data to the irregular graph-structured data. One typical GCN model can be defined as the following layer-wise scheme, i.e.,

$$H^{(l+1)} = \text{ReLU}(\tilde{D}^{-\frac{1}{2}} \tilde{A} \tilde{D}^{-\frac{1}{2}} H^{(l)} W^{(l)}), \tag{2}$$

where $H^{(l)} \in \mathbb{R}^{n \times d}$ is the hidden embedding matrix for the $l$-th layer, $W^l \in \mathbb{R}^{d \times d}$ is the trainable parameter matrix for the $l$-th layer, $\tilde{A} = A + I$ is the adjacency matrix associated with the self loop, $\tilde{D} = \sum_j \tilde{A}_{ij}$ is the degree matrix of $\tilde{A}$, and $H^{(l+1)}$ is the embedding matrix extracted for the next $l + 1$-th layer. The GCN model can effectively capture the global structural representations through the multiple stacked convolution layers defined by Eq.(2). Unfortunately, since the convolution operation depends on the node information propagation between adjacent nodes. The GNN suffers from
similar when the GCN becomes deeper, e.g., the GNN with more than $5$ convolution layers.

### 2.2 The Graph Auto-Encoder

The GAE is a powerful self-supervised framework for graph representation learning. Typical instances of GAEs include the DeepWalk [25] and the Node2Vec [7], where the encoders usually play an important role for learning the latent representations of vertices. By associating with the convolution operations of GNNs, the encoders of GAEs are able to cope with the non-Euclidean data [24]. As a self-supervised learning model, the aim of GAEs during the training process is the graph reconstruction [17], that can be categorized into the fine-grained and the coarse-grained targets.

Specifically, the fine-grained target contains either nodes or edges. For instance, the Variational Graph Auto-Encoder (VGAE) model [29] adopts two stages, i.e. the encoder and the decoder, to accomplish the representation learning. For an input graph $G(V, E)$, the goal of VGAE is to embed the graph through the encoder function $f : V \times E \rightarrow Z \in \mathbb{R}^{n \times d}$, that maps the original node feature matrix of the node set $V$ to the embedding matrix $Z$. Then, the decoder reconstructs the graph through the function $g : Z \rightarrow E'$, where $E'$ is the reconstructed edge set for the original edge set $E$. During the encoding and decoding processes, the VGAE obtains the conditional probability distributions

$q(Z \mid V, E)$ and $p(E' \mid Z)$ from the encoder and the decoder respectively, and the loss function is

$$\mathcal{L} = \text{KL}[q(Z \mid V, E) \| p(Z)] - \mathbb{E}_{q(Z|V,E)}[\log p(E' \mid Z)], \tag{3}$$

where $\text{KL}[\cdot]$ is the Kullback-Leibler divergence, $\mathbb{E}$ is the expection, and $p(Z)$ is the Gaussian prior.

On the other hand, the coarse-grained target contains the subgraph or the path structure. For example, the Heterogeneous Graph Masked Autoencoder (HGMA) [30] adopts the dynamic masking strategy to mask the nodes and the edges on the paths, and then reconstruct the path. Similar approaches also include the Masked Graph Autoencoder (MaskGAE) [14], that aims to reconstruct the masked edges and the node degrees jointly. Recently, some researchers have noted that the combination of the Graph Contrastive Learning (GCL) [38] and the GAE framework can further capture the complex interdependency residing on graphs. Under this scenario, the Self-supervised Learning for Graph Anomaly Detection (SL-GAD) [41] has been developed to obtain double subgraphs through the graph view sampling, and then respectively reconstructs them in two decoders for constrastive learning.

Although the above GAE methods significantly improve the performance of the graph representation learning, they still suffer from some common drawbacks. First, these GAE methods usually have superior performance for node classification, but have poor performance for graph classification. This is due to the fact that these methods only focus on the node feature reconstruction and ignore the global graph structure reconstruction, resulting in topological missing. Thus, these methods tends to be elusive for multiple downstream tasks [17]. Second, since the decoder and the encoder of these GAE methods usually employ the GNN models (i.e., the associated graph convolution operation) to extract the node feature information. The over-smoothing problem of GNNs mentioned in Section 2.1 also affects the feature learning for these GAE-based methods. Moreover, when the perturbation of the graph structure is conducted, the noise can also be propagated to the neighbor nodes through the edges. After several rounds of information passing, the generated graph representations may be noisy.

## 3 The Methodology

To overcome the aforementioned challenges, we propose a novel (**HC-GAE**) model to learn effective graph representations. The framework of the proposed model is shown in Figure 1. Similar to the classical GAEs, the HC-GAE model has two stages, including the encoder as well as the decoder. During the encoding process, the encoder can layer-wisely compress each sample graph into a series of coarsened graphs, and the results of the encoder can be employed as the graph-level representations for graph classification. During the decoding process, the decoder can reconstruct the structure of the original sample graph, and outputs the node-level representations for node classification. In the following subsections, we commence by introducing the encoder and decoder of the proposed HC-GAE model. Moreover, we define a family of loss function for the proposed HC-GAE model. Finally, we theoretically analyze the effectiveness of the proposed model.

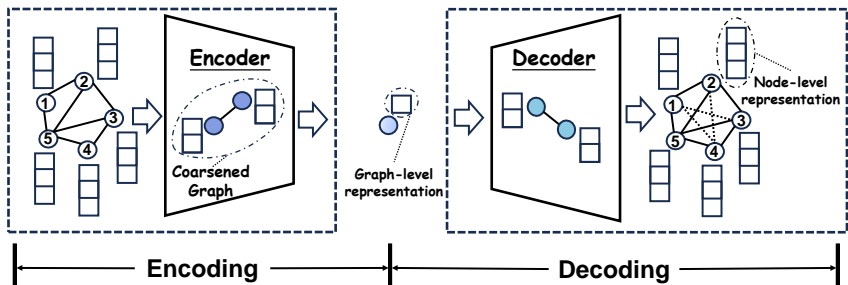

Figure 1: The architecture of our proposed HC-GAE model.

### 3.1 The GNN Encoder associated with the Hard Node Assignment

The first module of the proposed HC-GAE model is the encoder, that consists of multiple GNN layers and adopts the hierarchical computational architecture to compress the original input graph into a series coarsened graphs with shrinking sizes. Details of this GNN-based encoder are shown in Figure 2, where each layer includes the node assignment and coarsening processes. The first process is to generate separated subgraphs from the original graph, and the second process compresses the subgraphs into coarsened nodes of a coarsened graph.

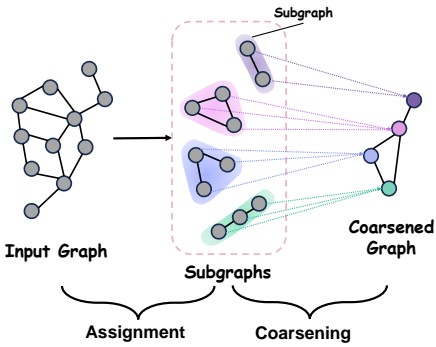

Figure 2: The computational architecture for our proposed layer in the GNN encoder.

**Hard Assignment.** For each $l$-th layer of the encoder, the input graph is denoted as $G^{(l)} = (X^{(l)}, A^{(l)})$, where $X^{(l)} \in \mathbb{R}^{n_{(l)} \times d_{(l)}}$ is the node feature matrix and $A^{(l)} \in \mathbb{R}^{n_{(l)} \times n_{(l)}}$ is the adjacent matrix. The number of nodes for $G^{(l)}$ is $n_{(l)}$, and each node has $d_{(l)}$ features. Note that, $G^{(l)}$ can be either the original input sample graph when $l = 1$ or the coarsened graph when $l > 1$. For the assignment process, the graph $G^{(l)}$ is decomposed into a number of separated subgraphs, by assigning the $n_{(l)}$ nodes into $n_{(l+1)}$ clusters through the hard node assignment. To achieve this, we commence by computing the soft node assignment matrix $S_{\mathrm{soft}}$ as

$$S_{\mathrm{soft}} = \begin{cases} \mathrm{softmax}(\mathrm{GNN}(X^{(l)}, A^{(l)})) & \text{if } l = 1 \\ \mathrm{softmax}(X^{(l)}) & \text{if } l > 1 \end{cases}, \tag{4}$$

where $S_{\mathrm{soft}} \in \mathbb{R}^{n_{(l)} \times n_{(l+1)}}$ and $n_{(l+1)} < n_{(l)}$. To guarantee the separability of the resulting subgraphs, we need to assign each node into an unique cluster through a hard node assignment matrix $S^{(l)} \in \{0, 1\}^{n_{(l)} \times n_{(l+1)}}$, where each $(i, j)$-th entry of $S^{(l)}$ can be computed through $S_{\mathrm{soft}}$, i.e.,

$$S^{(l)}(i, j) = \begin{cases} 1 & \text{if } S_{\mathrm{soft}}(i, j) = \max_{\forall j \in n_{l+1}} [S_{\mathrm{soft}}(i, :)] \\ 0 & \text{otherwise} \end{cases}. \tag{5}$$

Clearly, each $i$-th row of the hard assignment matrix $S^{(l)}$ selects the maximum element as 1 and the remaining elements as 0. As a result, the $i$-th node is only assigned to the $j$-th separated subgraph $G_j^{(l)}(V_j^{(l)}, E_j^{(l)})$, where $V_j^{(l)}$ consists of the nodes belonging to the $j$-th cluster and $E_j^{(l)}$ remains the original edge connections between the nodes in $V_j^{(l)}$ from the original input graph $G^{(l)}$.

**Coarsening.** The coarsening process compresses the separated subgraphs into coarsened nodes of the resulting coarsened graph. Given the feature and adjacency matrices $X_j^{(l)}$ and $A_j^{(l)}$ of the subgraph $G_j^{(l)}$, we first adopt a local graph convolution operation to extract the local structural information as

$$Z_j^{(l)} = A_j^{(l)} X_j^{(l)} W_j^{(l)}, \tag{6}$$

where $W_j^{(l)} \in \mathbb{R}^{d_{(l)} \times d_{(l+1)}}$ ($d_{(l)} > d_{(l+1)}$) is the trainable weight matrix of the $l$-th layer, and $Z_j^{(l)} \in \mathbb{R}^{|V_j^{(l)}| \times d_{(l+1)}}$ is the resulting local structural representation of $G_j^{(l)}$. To compress each subgraph $G_j^{(l)}$ into a a coarsened node, we define a mapping vector $\mathbf{s}_j^l$ as

$$\mathbf{s}_j^{(l)} = \mathrm{softmax}(A_j^{(l)} X_j^{(l)} D_j^{(l)}), \tag{7}$$

where $D_j^{(l)} \in \mathbb{R}^{d_{(l)} \times 1}$ is a trainable vector, and will play an important role in compressing each subgraph $G_j^{(l)}$ into the node of the coarsened graph. After several local graph convolution operations on the separated subgraphs in the $l$-th layer, we aggregate these local information to further generate the coarsened graph, as the input graph $G^{(l+1)}$ for the next $l + 1$-th layer. To this end, for the original input graph $G^{(l)}$, we collect the feature matrices $Z_j^{(l)}$ of all $j$-th subgraphs $G_j^{(l)}$ as the extracted node feature matrix $Z^{(l)} \in \mathbb{R}^{n_{(l)} \times d_{(l+1)}}$, whose node sequences follow the original graph $G^{(l)}$, i.e., each vertex feature vector of $Z^{(l)}$ is equal to that of the corresponding node in $G_j^{(l)}$. This is because each

node of $G_j^{(l)}$ essentially corresponds to an unique original node of $G^{(l)}$. Given the hard assignment matrix $S^{(l)}$ defined by Eq.(5) and the mapping vector $\mathbf{s}_j^{(l)}$ defined by Eq.(7), we compute the feature matrix $X^{(l+1)}$ and the adjacent matrix $A^{(l+1)}$ of the resulting coarsened graph $G^{(l+1)}$ as

$$X^{(l+1)} = \text{Reorder}\big[\mathop{\|}\limits_{j=1}^{n_{l+1}} \mathbf{s}_j^{(l)\top}\big]Z^{(l)}, \tag{8}$$

and

$$A^{(l+1)} = S^{(l)\top} A^{(l)} S^{(l)}, \tag{9}$$

where $\|$ is a concatenation operation, and the function $\text{Reorder}$ reorders the sequences of $[\cdot]$ to follow the same node sequence of $G_j^{(l)}$. For the encoder, we set the greatest layer number as $L$, and the resulting coarsened graph $G^{(L+1)} = (X^{(L+1)}, A^{(L+1)})$ is the input graph for the decoder. Furthermore, we adopt a non-parameterized readout function (e.g., the MaxPooling and MeanPooling) to further embed $X^{(L+1)}$ as ***the graph-level representation for graph classification***.

## 3.2 The GNN Decoder associated with the Soft Node Assignment

The second module of the proposed HC-GAE model is the decoder, that aims to reconstruct the structure of the original input sample graph, by expanding the each coarsened node into retrieved nodes probabilistically through the soft node assignment. Similar to the encoder, the decoder also consists of multiple GNN-based layers. Details of each decoder layer are shown in Figure 3, where the input is the retrieved graph and the output is the reconstructed graph.

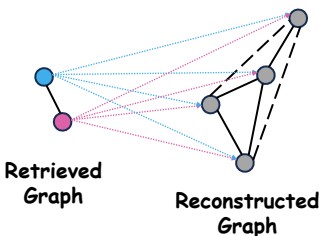

Figure 3: The illustration of our proposed layer in the GNN decoder.

**Reconstruction.** For each $l'$-th layer of the decoder, the input retrieved graph is denoted as $G'^{(l')} = (X'^{(l')}, A'^{(l')})$, where $X'^{(l')} \in \mathbb{R}^{n_{(l')} \times d_{(l')}}$ is the node feature matrix and $A'^{(l')} \in \mathbb{R}^{n_{(l')} \times n_{(l')}}$ is the adjacent matrix. Note that, when $l' = 1$, $G'^{(l')}$ is essentially the resulting coarsened graph $G^{(L)}$ of the encoder. For the reconstruction process with $G'^{(l')}$, we commence by computing the soft node assignment matrix $\bar{S}^{(l')} \in \mathbb{R}^{n_{(l')} \times n_{(l'+1)}}$ (i.e., the learnable node re-assignment matrix) as

$$\bar{S}^{(l')} = \text{softmax}(\text{GNN}_{l',\text{re}}(X'^{(l')}, A'^{(l')})). \tag{10}$$

Moreover, we further compute the node feature embedding of $G'^{(l')}$ as

$$\bar{Z}^{(l')} = \text{GNN}_{l',\text{emb}}(X'^{(l')}, A'^{(l')}). \tag{11}$$

Note that, the associated graph convolution operations $\text{GNN}_{l',\text{re}}$ and $\text{GNN}_{l',\text{emb}}$ for Eq.(10) and Eq.(11) do not share the parameters. Specifically, the operation $\text{GNN}_{l',\text{re}}$ generates a probabilistic distribution to expand the nodes of $G'^{(l')}$ to the corresponding nodes of the resulting reconstructed graph $G'^{(l'+1)}$, and the operation $\text{GNN}_{l',\text{emb}}$ extracts the new embeddings for $G'^{(l')}$. With $\bar{S}^{(l')}$ and $\bar{Z}^{(l')}$ to hand, we compute the node feature matrix $X'^{(l'+1)} \in \mathbb{R}^{n_{(l'+1)} \times d_{(l'+1)}}$ and the adjacency matrix $A'^{(l'+1)} \in \mathbb{R}^{n_{(l'+1)} \times n_{(l'+1)}}$ for the reconstructed graph $G'^{(l'+1)}$ as

$$X'^{(l'+1)} = \bar{S}^{(l')\top} \bar{Z}^{(l')}, \tag{12}$$

and

$$A'^{(l'+1)} = \bar{S}^{(l')\top} A'^{(l')} \bar{S}^{(l')}. \tag{13}$$

We employ $G'^{(l'+1)}$ as the input retrieved graph for the next $l' + 1$-th layer, and set the greatest layer number as $L'$ ($L'$ equals to $L$ of the encoder). The resulting graph $G'^{(L'+1)} = (X'^{(L'+1)}, A'^{(L'+1)})$ is the output of our HC-GAE, and $X'^{(L'+1)}$ is ***the node-level representations for node classification***.

### 3.3 The Loss Function

The proposed HC-GAE model consists of the encoder focusing on compressing the graph structural information and the decoder aiming at reconstructing the original graph structure. Therefore, the loss function $\mathcal{L}_{\mathrm{HC-GAE}}$ of HC-GAE includes two individual parts, and is defined as

$$
\begin{aligned}
\mathcal{L}_{\mathrm{local}} &= \sum_{l=1}^{L} \sum_{j=1}^{n_{(l+1)}} \mathrm{KL}[q(Z_j^{(l)} \mid X_j^{(l)}, A_j^{(l)}) \| p(Z^{(l)})], \\
\mathcal{L}_{\mathrm{global}} &= -\sum_{l=1}^{L} \mathbb{E}_{q(X^{(L)}, A^{(L)}) \mid X^{(l)}, A^{(l)})}[\log p(X'^{(L-l+2)}, A'^{(L-l+2)} \mid X^{(L)}, A^{(L)})], \\
\mathcal{L}_{\mathrm{HC-GAE}} &= \mathcal{L}_{\mathrm{local}} + \mathcal{L}_{\mathrm{global}},
\end{aligned}
\tag{14}
$$

where $\mathcal{L}_{\mathrm{local}}$ is the local loss, $\mathcal{L}_{\mathrm{global}}$ is the global loss, and $p(Z^{(l)})$ is the Gaussian prior for the $l$-th layer. Specifically, $\mathcal{L}_{\mathrm{local}}$ aims at training the subgraph generation, that reserves the local information and avoids the over-smoothing problem for the GNN-based encoder. $\mathcal{L}_{\mathrm{global}}$ focuses on reconstructing the graph features and structures. The combination of $\mathcal{L}_{\mathrm{local}}$ and $\mathcal{L}_{\mathrm{global}}$ broadens the reconstruction requirement for multiple downstream tasks (i.e., the node and graph classification), since $\mathcal{L}_{\mathrm{local}}$ is a regularization for the loss and can strengthen the graph representations associated with the additional local information for the proposed HC-GAE model. Note that, for the node classification, we only need to reconstruct the structure information (i.e., the adjacency matrix $A^{(\cdot)}$) rather than the node feature information (i.e., the feature matrix $X^{(\cdot)}$) for the loss function $\mathcal{L}_{\mathrm{HC-GAE}}$.

### 3.4 Discussions

**(a) Why is the proposed HC-GAE model effective for multiple downstream tasks?**

As we mentioned in Section 1 and 2.2, most existing GAE methods tend to over-emphasize the graph feature reconstruction, not only ignoring the topological structure information but also aggravating the over-fitting problem for the graph features [32]. Thus, these GAE methods are usually employed for node classification, and have poor performance for graph classification. To overcome this shortcoming, the proposed HC-GAE model adopt the following schemes. **First**, the HC-GAE adopts the node assignment strategy to improve the encoding and decoding performance. Specifically, for the encoder, we utilize the hard node assignment to decompose the input graph into separated subgraphs to extract local structural information, and this can reduce the redundancy for the graph-level representation learning. On the other hand, for the decoder, we use the soft node assignment to reconstruct the original graph structure, and employ the associated node features of the reconstructed graph as the node-level representations. The combination of these assignment strategies guarantees that the HC-GAE can effectively learn multi-level representations for various downstream tasks. **Second**, the loss function of the HC-GAE consists of both the local and global loss. The global loss $\mathcal{L}_{\mathrm{global}}$ can either reconstruct the graph features or the topological structures, reducing the over-emphasizing on graph features. The local loss $\mathcal{L}_{\mathrm{local}}$ not only plays a role of regularization in $\mathcal{L}_{\mathrm{HC-GAE}}$, but also captures the local information from the separated subgraphs, improving the generalization of the graph representations through the local information [39].

**(b) Why does the proposed HC-GAE model reduce the over-smoothing problem?**

Unlike the hierarchical-based GNN methods [29, 37], during the encoding process, we perform the local graph convolution operation within each individual separated subgraph. Thus, the information can not be propagated to other subgraphs, significantly reducing the over-smoothing problem.

## 4 Experiments

This section empirically evaluates the performance of the proposed HC-GAE[2] for both node and graph classification on benchmark datasets, whose detailed statistical information are shown in Table 1 and Table 2. Specifically, for the HC-GAE, we select the Adam optimizer to train the parameters, and set the epoch, hidden dimension and dropout as 50, 128 and 0.5, respectively. For the encoder and

---

[2]The source code is available on `https://github.com/JonathanGXu/HC-GAE`

Table 1: Datasets for node classification

| Datasets | Cora | CiteSeer | PubMed | Computers | CS |
|---|---|---|---|---|---|
| **Nodes** | 2708 | 3312 | 19717 | 13752 | 18333 |
| **Edges** | 5429 | 4660 | 44338 | 245861 | 81894 |
| **Features** | 1433 | 3703 | 500 | 767 | 6805 |
| **Classes** | 7 | 6 | 3 | 10 | 15 |

Table 2: Datasets for graph classification

| Datasets | IMDB-B | IMDB-M | PROTEINS | COLLAB | MUTAG |
|---|---|---|---|---|---|
| **Graphs** | 1000 | 1500 | 1113 | 5000 | 188 |
| **Nodes(mean)** | 19.77 | 13 | 39.06 | 74.49 | 17.93 |
| **Edges(mean)** | 96.53 | 65.94 | 72.82 | 2457.78 | 19.79 |
| **Classes** | 2 | 3 | 2 | 3 | 1 |

decoder of the HC-GAE, we set their greatest layer numbers $L$ and $L'$ as 3, and their node numbers for the 3 layers as $\{128, 64, 32\}$ and $\{32, 64, 128\}$. Moreover, we set the batch size and the learning rate as 1024 and $1e-2$ for node classification, and 64 and $5e-4$ for graph classification. For either the proposed HC-GAE or the alternative baselines, we adopt the 10-fold cross validation to compute classification accuracies, and follow the original setups for different baselines.

## 4.1 Node Classification

**Datasets and Baselines.** For node classification, we employ five real-world datasets, i.e., Cora, CiteSeer, PubMed, Amazon-Computers and Coauthor-CS. Moreover, we compare the proposed model with six classical self-supervised models, i.e., DGI [33], VGAE [29], SSL-GCN [42], GraphSage [8], GraphMAE [9], and S2GAE [28]. To fairly compare the proposed model and other baselines, we follow the previous study [10] to carry out the related experiments, and utilize the SVM classifier to predict the node labels, and the classification accuracies are shown in Table 3.

**Results.** Clearly, the proposed HC-GAE outperforms most of the baselines. Only the accuracy of the GraphMAE is a little higher than our model. The reasons for the effectiveness are twofold. First, the HC-GAE model is defined based on a bidirectionally hierarchical computational architecture, and can extract more effective node feature information to compress and reconstruct the original sample graph structures. Second, the HC-GAE can significantly reduce the over-smoothing problem.

Table 3: Node classification performance based on accuracy. A.R. is the average rank.

| Datasets | Cora | CiteSeer | PubMed | Computers | CS | A.R. |
|---|---|---|---|---|---|---|
| DGI | 85.41±0.34 | 74.51±0.51 | 85.95±0.66 | 84.68±0.39 | 91.33±0.30 | 4.0 |
| VGAE | 83.60±0.52 | 63.37±1.21 | 78.23±1.63 | 87.21±0.26 | 89.79±0.09 | 5.2 |
| SSL-GCN | 57.29±0.13 | 59.57±1.77 | 75.06±0.37 | 80.49±0.10 | 84.71±0.95 | 6.8 |
| GraphSage | 74.30±1.84 | 60.20±2.15 | 81.96±0.74 | 87.05±0.25 | 89.74±0.19 | 5.6 |
| GraphMAE | 85.45±0.40 | 72.48±0.77 | 85.74±0.14 | 88.04±0.61 | **93.47±0.04** | 3.0 |
| S2GAE | 86.15±0.25 | 74.60±0.06 | 86.91±0.28 | 90.94±0.08 | 91.70±0.08 | 2.2 |
| **HC-GAE** | **87.97±0.10** | **75.29±0.09** | **87.56±0.35** | **91.07±0.14** | 92.28±0.07 | **1.2** |

## 4.2 Graph Classification

**Datasets and Baselines.** For graph classification, we adopt five standard graph datasets, i.e., IMDB-B, IMDB-M, PROTEINS, COLLAB and MUTAG. Moreover, we compare the proposed model with a graph kernel (i.e., WLSK [27]), two supervised GNN models (i.e., DGCNN [40] and D-iffPool [37]), and four self-supervised GAE models (i.e., Graph2Vec [23], InfoGCL [36], Graph-MAE [9], S2GAE [28]). We employ the SVM associated with the resulting graph representations to compute the classification accuracies in Table 4, following the previous study in [9].

**Results.** Clearly, the proposed HC-GAE model outperforms most of the alternative baselines. Only accuracy of the S2GAE on the COLLAB dataset is a little higher than the proposed HC-GAE. The effectiveness of the HC-GAE are due to the following reasons. First, the alternative GAE-based models (e.g., the GraphMAE) focuses on the graph feature reconstruction rather than the topological structure information. Thus these models are only effective for node classification and have poor performance for graph classification. By contrast, the HC-GAE can simultaneously focus on either the node feature or the structure information. Second, most of the alternative deep learning baselines are essentially GNN-based methods, that usually suffers from the over-smoothing problem. By contrast, the proposed HC-GAE can significantly reduce the over-smoothing through the local graph convolution within separated subgraphs of limited sizes. Third, unlike the proposed HC-GAE that is

a deep learning method, the alternative WLSK kernel cannot provide an end-to-end learning manner, i.e., the kernel computation is separated from the training of the SVM classifier.

Table 4: Graph classification performance based on accuracy. A.R. is the average rank.

| Datasets | IMDB-B | IMDB-M | PROTEINS | COLLAB | MUTAG | A. R. |
|---|---|---|---|---|---|---|
| WLSK | 64.48±0.90 | 43.38±0.75 | 71.70±0.67 | N/A | 80.72±3.00 | 7.75 |
| DGCNN | 67.45±0.83 | 46.33±0.73 | 73.21±0.34 | N/A | 85.83±1.66 | 6.25 |
| DiffPool | 72.6±3.9 | 47.2±1.8 | 75.1±3.5 | 78.9±2.3 | 85.0±10.3 | 5.20 |
| Graph2Vec | 71.10±0.54 | 50.44±0.87 | 73.30±2.05 | N/A | 83.15±9.25 | 5.75 |
| InfoGCL | 75.10±0.90 | 51.40±0.80 | N/A | 80.00±1.30 | 91.20±1.30 | 3.50 |
| GraphMAE | 75.52±0.66 | 51.63±0.52 | 75.30±0.39 | 80.32±0.46 | 88.19±1.26 | 3.20 |
| S2GAE | 75.76±0.62 | 51.79±0.36 | 76.37±0.43 | **81.02±0.53** | 88.26±0.76 | 2.00 |
| **HC-GAE** | **76.72±0.60** | **51.90±1.47** | **78.13±1.37** | 80.41±0.02 | **92.38±1.17** | **1.20** |

## 4.3 Ablation Study

To analyze the effectiveness of the proposed HC-GAE one step further, we replace the the hard assignment strategy of the encoder with the soft assignment strategy, and perform the same setups for graph classification. We denoted the proposed model with the soft assignment as **HC-GAE-SE**. Under this scenario, the encoder of the HC-GAE-SE cannot decompose the graph into separated subgraphs and cannot employ the local graph convolution operation to extract the feature information, since each node will be assigned to all clusters with different weights. The experimental comparisons are shown in Figure 4. Obviously, the performance of the HC-GAE-SE is lower than the proposed HC-GAE on any dataset. The reasons are twofold. First, unlike the proposed HC-GAE, the HC-GAE-SE cannot avoid the influence of the over-smoothing problem, without the local convolution within the separated subgraphs. Second, the loss function of the proposed HC-GAE relies on the local graph information through the separated subgraphs. However, this is not available for the HC-GAE-SE, thus it cannot extract the local information to enhance the effectiveness.

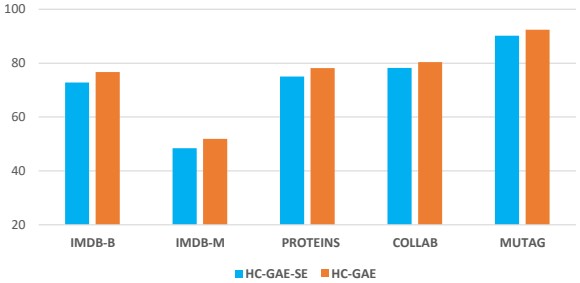

Figure 4: The ablation experiments on graph classification task.

## 5 Conclusions

In this paper, we have proposed a novel HC-GAE model to learn effective structural characteristics for either graph classification or node classification. During the encoding process, the HC-GAE has adopted the hard node assignment to decompose the input graph into a family of separated subgraphs, and compressed these subgraphs into coarsened nodes, resulting in a coarsened graph. During the decoding process, the HC-GAE has employed the soft node assignment to reconstruct the original graph structure, by expanding the coarsened graph of the encoder. The proposed HC-GAE can not only extract hierarchical structural features of the original graph, but also reduce the notorious over-smoothing problem arising in most classical GAEs, by restricting the convolution operation within each separated subgraphs during the encoding process. Experiments have demonstrated that the proposed HC-GAE has superior performance for both node and graph classification.

Our future work is to employ the structurally aligned (sub)structures, that have been successfully explored in our previous works [4, 5] for the proposed HC-GAE model. This will naturally result in new Transitive-Aligned HC-GAE models to guarantee the consistency between the structural correspondence information as well as the spatial position over all graphs. Finally, we will also use the Perron-Frobenius operator of hypergraphs [1, 3] associated with the proposed HC-AGE model, resulting in novel hypergraph-based HC-GAE models.

## Acknowledgments

This work is supported by the National Natural Science Foundation of China under Grants T2122020, 61602535 and 61976235. Ming Li acknowledged the supports from the National Natural Science Foundation of China (No. U21A20473, No. 62172370) and the Jinhua Science and Technology Plan (No. 2023-3-003a). This work is also supported in part by the Program for Innovation Research in the Central University of Finance and Economics, and the Emerging Interdisciplinary Project of CUFE.

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
