# OpenReview forum: "HC-GAE: The Hierarchical Cluster-based Graph Auto-Encoder for Graph Representation Learning"
_NeurIPS.cc/2024/Conference — NeurIPS 2024 poster_

### Official Review · Reviewer_M4BE · 2024-06-21

**Soundness:** 1
**Presentation:** 3
**Contribution:** 1
**Rating:** 3
**Confidence:** 4

**Summary:**

This paper proposes a novel Hierarchical Cluster-based Graph Auto-encoder (HC-GAE) for unsupervised graph representation learning. HC-GAE can reduce the over-smoothing problem and generalize to multiple downstream tasks.

**Strengths:**

1. The motivation is clear and easy to understand.
2. The proposed method seems to be simple and effective.
3. This paper is easy to follow.

**Weaknesses:**

1. The novelty is limited. (1) Both the motivations are not novel. Over-smoothing is a classic problem on GNNs and has been discussed for many years. Multi-task ability has also been discussed in previous literature [1], which has much better multi-task performance than the proposed method. (2) The method is not very novel either. The method is just a combination of existing methods (DiffPool, VGAE) with simple modification.
2. One of the motivations of this paper is to enhance the performance of GAEs for multiple downstream tasks, and the authors also emphasize it for the proposed method. Thus the authors are encouraged to conduct experiments on link prediction task, which is also very important in graph learning and very different from node classification and graph classification. If HC-GAE  is not designed for this task or cannot perform well on it, the authors should discuss the limitation on downstream tasks and highlight the discussion.
3. The loss function requires to compute multiple KL-divergence, and may require a lot of time to compute. The authors are encouraged to analyze the time and space complexity, and also conduct experiments to show the computational cost.
4. The baselines are not enough and kind of old. The authors are encouraged to include more recent baselines, including auto-encoder based methods (e.g., GraphMAE2 [2]) and contrastive learning based methods (e.g., CCA-SSG [3], GGD [4] and [5]).
5. The authors are also encouraged to include experimental results on more datasets.
6. There is no sensitivity test on the hyper-parameters.

[1] Multi-task Self-supervised Graph Neural Networks Enable Stronger Task Generalization. In ICLR 2023.

[2] GraphMAE2: A Decoding-Enhanced Masked Self-Supervised Graph Learner. In WWW 2023.

[3] From Canonical Correlation Analysis to Self-supervised Graph Neural Networks. In NeurIPS 2021.

[4] Rethinking and Scaling Up Graph Contrastive Learning: An Extremely Efficient Approach with Group Discrimination. In NeurIPS 2022

[5] SEGA: structural entropy guided anchor view for graph contrastive learning. In ICML 2023.

**Questions:**

Please refer to "Weaknesses".

**Limitations:**

Please refer to "Weakness 2".

---

> ### Author Rebuttal · Authors · 2024-08-07
>
> Q1: Both the motivations are not novel. Over-smoothing is a classic problem and Multi-task ability has also been discussed. The authors are encouraged to conduct experiments on link prediction task, and discuss the limitation on downstream tasks.
>
> A1: We would like to further explain and emphasize the motivation and the contribution of this work. First of all, the reviewer is correct that the over-smoothing problem is a very classical problem. But it still seriously influences most of the GNN-based methods by now, so that it has been discussed for many years until now, indicating that the over-smoothing is always an important issue for the development of novel GNN-based methods.
>
> On the other hand, the DiffPool and the GAE are both popular methods after 2018. Specifically, the DiffPool can be seen as a kind of Hierarchical Pooling methods that rely on constructing the original graph as the hierarchical pyramid-like structures, for the purpose of extracting meaningful structural characteristics of the original graph. These hierarchical structures are formed by gradually compressing the nodes into a number of corresponding clusters based on the node features, that are extracted through the GNN model, i.e., the node features are computed by performing the graph convolution operation. The GAE also needs to extract the node features through the graph convolution operation for either the decoder and encoder. Since, the required graph operations for both the DiffPool and VGAE models suffer from the over-smoothing problem, these two classical methods still have theoretical drawbacks.
>
> To overcome the above drawbacks, the novel contributions of this work are twofold. First, unlike the DiffPool that relies on the graph operation over the global graph structure, the proposed HC-GAE is based on the new subgraph convolution framework. Specifically, we propose to employ the hard node assignment to assign the nodes into separated clusters, resulting in separated subgraphs. The convolution operation is performed within each individual subgraphs, and the node information cannot be propagated between different subgraphs, this can thus significantly reduce the over-smoothing problem and improve the performance. Second, unlike some classical GAE-based methods (also see R#3-Q2&A2), the proposed HC-GAE can simultaneously reconstruct both the topological structures and the node features through the hierarchical strategy associated with the node assignment operation, i.e., the hard node assignment for the encoder and the soft node assignment for the decoder. Moreover, unlike some Hierarchical GAE methods (also see R#3-Q2&A2) that essentially rely on the node drop pooling (i.e., the masking strategy) and focus more on the node feature reconstruction, the proposed HC-GAE will not drop any node information, reducing the topological information loss, effectively extracting bidirectionally hierarchical structural features of the original sample graph.
>
> In summary, both the proposed HC-GAE and the DiffPool belong to the kind of the hierarchical strategy-based GNN methods, but they are still theoretically different in terms of the detailed definition and motivation. Moreover, the proposed HC-GAE is also different from the classical GAE methods, and can significantly overcome their drawbacks. We will follow the review’s suggestion and add these statement in the manuscript, making the manuscript more polish.
>
> For the link prediction problem, we thank the reviewer’s constructive suggestion. This enlightens our future work, and we will discuss this in the conclusion as future works.
>
> Q2: The authors are encouraged to analyze the time and space complexity.
>
> A2: Thanks for the suggestion. We have briefly analyzed the complexity as follows.
>
> For the encoding and decoding process, given the input graph with node set $V$ and edge set $E$, the proposed model requires the storage complexity of $\mathcal{O}(\frac{1}{\mu}\|V\|^2)$ where each cluster has $\mu$ nodes. By contrast, the DiffPool needs $\mathcal{O}(\|V\|^2)$. For the time complexity, if we set the layer as $k$, the proposed model requires $\mathcal{O}(k\mu\log\frac{\|V\|^2}{\mu})$. Clearly, the proposed model has the similar time and space complexity with the DiffPool.
>
> Furthermore, we will following the reviewer’s suggestion and generate a family toy sample graphs with increasing sizes (e.g., from 50, 100, 150 t0 5000), and evaluate how time and space complexity varies with the increasing sizes, demonstrating the efficiency.
>
> Q3: The baselines are not enough and kind of old. The authors are encouraged to include more recent baselines, including auto-encoder based methods.
>
> A3: Thank you for the constructive suggestion. In fact, for the current experiments, we have compared the proposed model with some methods published in recent 3 years. For the node classification, these methods include: S2GAE (WSDM 2023), GraphMAE (KDD 2022). For the graph classification, these methods include: S2GAE (WSDM 2023), GraphMAE (KDD 2022), InfoGCL (NIPS 2021). To make the experiments more convincing, we have also compared the proposed method with the newest methods published in 2024, and the results are shown as follows. More experiments will be evaluated during the review stage, and please feel free to discuss with us for the experiment. Finally, we will also do some sensitivity test on the hyper-parameters, following the reviewer’s suggestion.
>
> Table 1. The experiments for node classification. For the datasets Cora, CiteSeer and PubMed, the results for Graph U-Nets (2024) are 84.4 ± 0.60, 73.2 ± 0.50, and 79.6 ± 0.20; for Hi-GMAE (2024) are 86.4 ± 0.50, 74.5 ± 0.80, and 85.3 ± 0.20; for Ours (HC-GAE)	are 88.0 ± 0.10, 75.3 ± 0.10, and 87.6 ± 0.40.
>
> Table 2. The experiments for graph classification. For the datasets PROTEINS and COLLAB, the results for Graph U-Nets (2024) are 77.68 and	77.56; for Hi-GMAE (2024) are 76.63 and 82.16; for ours (HC-GAE) are 78.13 and 80.41.

---

> ### Author Response · Authors · 2024-08-09
> **About out responses**
>
> Dear Reviewers,
>
> Thanks for your efforts and the constructive suggestions for this paper. We have provided the response based on your comments.
>
> Please read our responses and feel free to discuss with us during the reviewer-author discussion stage, if you have any concerns for our response.
>
> Best Regards,
>
> The authors

---

> ### Comment · Reviewer_M4BE · 2024-08-12
>
> Thanks for the author's response, but my concerns are not addressed.

---

> ### Comment · Reviewer_M4BE · 2024-08-12
>
> After reading the rebuttal, I feel that my concerns are not addressed.
> 1. I have listed 6 weak points in the review. However, the authors only give responses to 3 points.
> 2. I have listed some baselines, including references, but the authors did not use them. Those baselines are very fast to run.
>
> Besides, the authors seem to be not familiar with this field
> 1. I still cannot understand the novelty. This method is just a combination of existing techniques.
>     1. There have been some surveys on Graph Pooling for Graph Neural Networks [1].
>     2. "Hard node assignment" has been widely used in graph coarsening [2, 3, 4, 5, 6, 7, 9].
>     3. "Simultaneously reconstruct both the topological structures and the node features" has been used in [7, 8, 10].
> 2. The complexity analysis is wrong. In graph neural networks, the adjacency matrices are implemented with sparse matrices, in order to reduce the complexity. Thus the complexity of the DiffPool is $O(|E| + |V| n_1)$ [7], where $|E|$ is the number of edges in the graph, $|V|$ is the number of nodes in the graph, and $n_1$ is the number of nodes after the first coarsening operation.
>
>
> ---
>
> [1] Graph Pooling for Graph Neural Networks: Progress, Challenges, and
> Opportunities. In IJCAI 2023.
>
> [2] Training Large-Scale Graph Neural Networks Via Graph Plartial Pooling. IEEE Transactions on Big Data.
>
> [3] Efficient Representation Learning of Subgraphs by Subgraph-To-Node Translation. ICLR 2022 workshop.
>
> [4] CC-GNN:A Community and Contraction-based Graph Neural Network. In ICDM.
>
> [5] SizeShiftReg: a Regularization Method for Improving Size-Generalization in Graph Neural Networks. NeurIPS 2022.
>
> [6] SMGRL: A Scalable Multi-resolution Graph Representation Learning Framework. Arxiv
>
> [7] GraphZoom: A multi-level spectral approach for accurate and scalable graph embedding. In ICLR 2020.
>
> [8] HARP: Hierarchical representation learning for networks. In AAAI 2018.
>
> [9] Spectral Clustering with Graph Neural Networks for Graph Pooling. In ICML 2020.
>
> [10] Graph U-Nets. In ICML 2019.

---

> ### Author Response · Authors · 2024-08-13
> **Replying to About out responses**
>
> Thank you for the responses as well as the constructive suggestions.
>
> After reading the new response from the review, we feel very strange and startled that the reviewer thinks he is not respected during the rebuttal stage. Maybe, there are some unnecessary misunderstandings between us, and we would like to explain in details and again show our respects to the reviewer.
>
> Please note, this does not mean that we entreat the acceptance for this paper. In fact, we also know this work still has some weaknesses, since this is the first work of a student who will start his PhD from this September. The other authors have published many top conferences and journals including nearly 30 TPAMI papers, thus we clearly understand the concerns raised by the reviewer, especially for the TOP conference like NIPS. By contrast, we just hope that the rebuttals can really help the student realize the drawbacks of his work, and know how to make this paper more polish in future.
>
> First, we have expressed our appreciation for the reviewer's efforts on reviewing this paper. We also thanked the reviewer's suggestions at the beginning of each response. As summary, we also said that "please feel free to discuss with us". Thus, we trust that we have expressed our sincere respects to the reviewer.
>
> Second, we have tried our best to answer each of the reviewer's concerns. Because some of the answers simultaneously correspond to more than one point, it just seems that there are only three answers. Specifically, A1 corresponds to the points 1 and 2 raised by the reviewer, A2 corresponds to the point 3 raised by the reviewer, and A3 corresponds the points 4-6 raised by the reviewer, respectively.
>
> Third, as explained in the rebuttal, the aim of this paper is not only to utilize the hard node assignment. Moreover, the hard node assignment is not only used to construct the coarsened nodes. More specifically, we employ the hard node assignment to generate a number of separated subgraphs, and perform the subgraph convolution operation to interdict the node information propagation between different subgraphs, reducing the over-smoothing problem. As a result, this is entirely different from most of the existing works.
>
> Fourth, thanks for correcting the mistake of the time complexity, and giving these new references. These will enlighten the student's future works.
>
> Finally, we thank the reviewer's new suggestions, and we have given our above responses to the new concerns raised by the reviewer. We hope our new response can eliminate the misunderstanding between us, and please trust that we are always appreciated with the reviewer's suggestions.
>
> Thank you very much again for the responses.
>
> Best Regards,
>
> The authors

---

### Official Review · Reviewer_Udb8 · 2024-07-10

**Soundness:** 2
**Presentation:** 3
**Contribution:** 2
**Rating:** 4
**Confidence:** 4

**Summary:**

The paper presents a Hierarchical Cluster-based Graph Auto-Encoder (HC-GAE) for improved graph representation learning. HC-GAE uses hard node assignment for encoding and soft node assignment for decoding; thus, it enables hierarchical compression and expansion of graphs. The authors argue their method can reduce the over-smoothing effect by limiting message-passing to individual subgraphs. HC-GAE effectively improves performance in node and graph classification tasks on real-world datasets.

**Strengths:**

The paper is well-written and includes all the necessary details. The proposed model solves a significant problem in the graph representation community and numerical results are promising.

**Weaknesses:**

The code should be released at the review stage to check reproducibility. Especially for empirical work, releasing codes is a prerequisite for acceptance to me.

The proposed HC-GAE is similar to existing models that employ graph auto-encoders with hierarchical (multi-level, multiresolution) pooling (or coarsening). The authors should include a discussion comparing HC-GAE with these models, both qualitatively and quantitatively. Without this discussion, the paper's claim of novelty cannot be highlighted.
1. Graph U-Nets
1. Mesh Variational Autoencoders with Edge Contraction Pooling
1. Masked Graph Auto-Encoder Constrained Graph Pooling
1. GRAPH AUTOENCODER FOR GRAPH COMPRESSION AND REPRESENTATION LEARNING
1. Multiresolution equivariant graph variational autoencoder
1. Hi-GMAE: Hierarchical Graph Masked Autoencoders

It is unclear whether the over-smoothing effect does not exist in HC-GAE truly, as there are no experimental results against the number of layers. If GNN is applied to the coarsened graph, message-passing between coarsened nodes will occur, potentially leading to over-smoothing. To support their argument, the authors should empirically validate the claim that HC-GAE reduces over-smoothing (i.e., how performance varies by the number of layers).

**Questions:**

- I think “compressing procedure during the decoding process (line 9 – 10)” should be “compressing procedure during the encoding process”. Is it a typo?

---

> ### Author Rebuttal · Authors · 2024-08-07
>
> Q1:The code should be released at the review stage to check reproducibility. Especially for empirical work, releasing codes is a prerequisite for acceptance to me.
>
> A1: Thanks for the constructive suggestion. We have provided a demo to the reviewer, please see the official comment with the anonymized link, and this follows the request of policy of the Reviewing/Discussion process. The reviewer can also use other standard datasets we raised in the manuscript with our code. We trust that this can help the reviewer demonstrate the reproducibility for the proposed method. Please feel free to discuss with us for the code during the discussion stage. Moreover, we promise that we will also release our code on the Github after the review process.
>
> Q2: The proposed HC-GAE is similar to existing models that employ graph auto-encoders with hierarchical (multi-level, multiresolution) pooling (or coarsening). The authors should include a discussion comparing HC-GAE with these models, both qualitatively and quantitatively. Without this discussion, the paper's claim of novelty cannot be highlighted. E.g., 1) Graph U-Nets; 2) Mesh Variational Autoencoders with Edge Contraction Pooling; 3) Masked Graph Auto-Encoder Constrained Graph Pooling; 4) GRAPH AUTOENCODER FOR GRAPH COMPRESSION AND REPRESENTATION LEARNING; 5) Multiresolution equivariant graph variational autoencoder; 6) Hi-GMAE: Hierarchical Graph Masked Autoencoders
>
>
> A2：Thanks for the constructive suggestion. Although some existing Hierarchical GAE methods also employ the auto-encoder with the hierarchical pooling, there are still some important theoretical differences between the proposed HC-GAE and these alternative methods.
>
> First, unlike these alternative Hierarchical GAE methods, the proposed HC-GAE not only hierarchically constructs the series of coarsened graphs with shrinking sizes, but is also defined by associating with the separated subgraph convolution operations during the hierarchical pooling process. Thus, unlike these alternative methods relaying on the global convolution over the whole graph structure, the proposed HC-GAE can naturally restrict the information propagation in each individual separated subgraph, significantly reducing the over-smoothing problem. Second, the alternative Hierarchical GAE methods are essentially based on the node drop pooling (i.e., the masking strategy), that focuses more on the node feature reconstruction, resulting in topological missing and weakening the structure information reconstruction. By contrast, the proposed HC-GAE needs to simultaneously reconstruct both the topological structures and the node features through the hierarchical pooling strategy associated with the node assignment operation. Thus, the proposed HC-GAE can effectively extract bidirectionally hierarchical structural features of the original sample graph, in terms of either the adjacency matrices (i.e., the topological information) or the node feature matrix (i.e., the node representation). We will cite the references suggested by the reviewer, and add the above discussions in the manuscript, making the paper more polish.
>
> Furthermore, to empirically demonstrate the effectiveness, we will also compare the proposed HC-GAE model with the alternative Hierarchical GAE suggested by the reviewer. As a preliminary evaluation, we have collected several models suggested by the reviewers, the results are shown as follows and our method can significantly outperform the alternative methods. Note that, more experiments are performing now, and we will add the new experimental results as well as the above theoretical discussions in the final manuscript, making this paper more polish.
>
> Table 1. The experimental results (node classification); the datasets are Cora, CiteSeer, and PubMed.
> The results for Graph U-Nets	are 84.4 ± 0.6, 73.2 ± 0.5, and 79.6 ± 0.2;
> for Hi-GMAE	are 86.4 ± 0.5, 74.5 ± 0.8, and 85.3 ± 0.2;
> for ours(HC-GAE) are 88.0±0.1, 75.3±0.1, 87.6 ± 0.4.
>
> Table 2. The experimental results (graph classification), the datasets are PROTEINS and COLLAB.
> The results for Graph U-Nets	are 77.68	and 77.56;
> for Hi-GMAE are 76.63 and 82.16;
> for ours(HC-GAE)	are 78.13	and 80.41.
>
> Q3: It is unclear whether the over-smoothing effect does not exist in HC-GAE truly, as there are no experimental results against the number of layers. If GNN is applied to the coarsened graph, message-passing between coarsened nodes will occur, potentially leading to over-smoothing. To support their argument, the authors should empirically validate the claim that HC-GAE reduces over-smoothing (i.e., how performance varies by the number of layers).
>
> A3: Thanks for the constructive suggestion. Through the theoretical viewpoint, the definition of the  proposed HC-GAE associated the separated subgraph convolution operation, and the node information can not be propagated to other individual subgraphs, significantly reducing the over-smoothing problem. For the empirical viewpoint, we have select three alternative methods that we have evaluated and suggested by the reviewer, we evaluate how the classification accuracies vary with number of layers. We find that the proposed HC-GAE can significantly outperform these alternative methods, especially with more than 3 layers for either the encoder or the decoder. We will add these new results in the final manuscript, making the experiment and the statement more self-contained.

---

> > ### Comment · Reviewer_Udb8 · 2024-08-12
> >
> > Thank you for opening the codes. I will raise my score to 4.
> >
> > However, I cannot still find a clear difference between the proposed and existing models.
> >
> > About Q&A2-1. Separate subgraph convolution: If the separated subgraph convolution is the first novel contribution that makes a difference, why does the paper focus only on the auto-encoder? Why do the authors not apply this operation for the general GNN layers to reduce the over-smoothing effect? (e.g., separated subgraph GCN, separated subgraph GAT, ...). I cannot find a good reason why this paper is positioned as it is now.
> >
> > About Q&A2-2. Node assignment: From what I understand from the responses and paper description so far, it is difficult to see clear contributions other than a 'combination' of hard and soft assignments. I think existing pool and un-pool methods have used at least one kind of operation in this paper.

---

> > > ### Author Response · Authors · 2024-08-13
> > >
> > > Thanks for your appreciation. We will update our paper, following the reviewer’s suggestions.
> > >
> > > About Q&A2-1.
> > > We think that the separated subgraph convolution has great potential for solving the over-smoothing problem. In future work, we will apply it to improve the GNN layers. Thanks for your advice about the separated subgraph convolution. It helps us recognize the worth of the proposed convolution operation. In this paper, we focus more on the graph representation learning and the multiple downstream tasks.
> > >
> > > About Q&A2-2.
> > > Node assignment is one of the key contributions mentioned in our paper, but not all. To explain in detail, we summarize our contributions as follows.
> > > The first contribution of the proposed HC-GAE model is that it performs the subgraph convolution operation in each individual subgraph. Compared to the other hierarchical GAEs, our subgraph generation reduces the information propagation through edges, which is the main factor of the over-smoothing. On the other hand, we adopt the soft node assignment to reconstruct the original graph structure in the decoding. And the outputs of the decoder can be employed as the node-level representations. Since the HC-GAE is based on the hierarchical framework, it can effectively extract bidirectionally hierarchical structural features, training for multiple downstream tasks.
> > > Second, we re-design the loss function to be suitable for our model on multiple downstream tasks. Since the HC-GAE generates the bidirectionally hierarchical structural features beyond the original GAEs, the traditional loss function might neglect their rich semantics in training. The existing hierarchical pool methods or un-pool methods hardly discuss the loss related to the bidirectionally hierarchical features.
> > > Last but not least, we realize the improvement of the hierarchical GAE performance. The results show that our model effectively learns the graph representations for multiple downstream tasks.

---

> ### Author Response · Authors · 2024-08-07
> **The anonymized link for the code requested by the reviewer**
>
> The reviewers asked for code during the rebuttal stage, below is the anonymized link for the code.
>
> https://anonymous.4open.science/r/HC-GAE-ECD7

---

> ### Author Response · Authors · 2024-08-09
> **About our responses and the updated anonymous link for the CODE**
>
> Dear Reviewers,
>
> Thanks for your efforts and the constructive suggestions for this paper. We have provided the response based on your comments.
>
> Moreover, we have updated the code requested by the reviewer, please see the anonymous link: https://anonymous.4open.science/r/SSHPool-FB16.
>
> Please read our responses and feel free to discuss with us during the reviewer-author discussion stage, if you have any concerns for our response.
>
> Best Regards,
>
> The authors

---

### Official Review · Reviewer_94wD · 2024-07-12

**Soundness:** 3
**Presentation:** 3
**Contribution:** 4
**Rating:** 8
**Confidence:** 4

**Summary:**

This paper develops a novel GAEs, namely the Hierarchical Cluster-based GAE (HC-GAE) model, to learn effective features for either node classification or graph classification. To extract the  bidirectionally hierarchical structural features of the original graph, this paper first utilize the hard node assignment to transform the original graph into a family of coarsened graphs, and then utilize the soft assignment to reconstruct the original graph. During the encoding process the convolution operation is restricted within each separated subgraph, so this HC-GAE can address the shortcoming of over-smoothing problem. This new model has superior performance on both node classification or graph classification tasks.

**Strengths:**

1.The idea of this work is interesting, and the bidirectionally hierarchical structure learning based on the adaptive node assignment seems novel to me.
2.The proposed HC-GAE model is flexible for either node classification or graph classification.
3.The paper is clearly written and easy to follow, and the experimental results also demonstrate the effectiveness of the new HC-GAE model.

**Weaknesses:**

Although this paper introduces a novel graph representation learning method, but some problems still need to be addressed or be clearer.
1.Why the HC-GAE model utilizes the hard and soft assignment for the encoder and decoder respectively? As I see, the author can only use any of these assignment strategies, right?
2.If the hard assignment can help the new model to reduce the over-smoothing problem, how about the soft-assignment?
3.Why the reconstructed features are effective for node classification? Why do not use the middle features?
4.Although this paper is clearly written, but the writing of Section 3 can be more polished.

**Questions:**

See the weakness raised above.

**Limitations:**

yes

---

> ### Author Rebuttal · Authors · 2024-08-07
>
> Although this paper introduces a novel graph representation learning method, but some problems still need to be addressed or be clearer.
>
> Q1: Why the HC-GAE model utilizes the hard and soft assignment for the encoder and decoder respectively? As I see, the author can only use any of these assignment strategies, right?
> A1: One key innovation of the proposed HC-GAE model is that it performs the subgraph convolution operation in each individual subgraph, so that the node information can only be propagated in a separated subgraph, significantly reducing the over-smoothing problem. To eliminate the structural connection between different subgraphs, each node of the original input sample graph need to be assigned to an unique cluster. The soft assignment will assign each node to multiple clusters associated with different probabilities, thus it cannot help us to reduce the over-smoothing problem.
>
>
> Q2: If the hard assignment can help the new model to reduce the over-smoothing problem, how about the soft-assignment?
> A2: Thanks for the suggestion. The reason is similar to the above problem. The soft assignment cannot guarantee that each node will be assigned into the same cluster, so that one node information may be existed in all other subgraphs, so that we cannot restrict the node information propagate in an individual separated subgraph. In other words, each node can still propagate its information to all other nodes. With the model architecture becomes deeper, the over-smoothing problem will appear.
>
> Q3: Why the reconstructed features are effective for node classification? Why do not use the middle features?
> A3: Sorry that we didn’t explain this in details. The proposed HC-GAE model is one kind of the hierarchical structure-based GNN methods, that can hierarchically constructs the series of coarsened graphs with shrinking sizes. Thus, the size of the coarsened graph in the middle layer is smaller than the original graph, i.e., many nodes of the original graph are compressed as the coarsened node. Clearly, the coarsened nodes are not suitable for the classification of original nodes.
>
>
> Q4: Although this paper is clearly written, but the writing of Section 3 can be more polished.
> A4: Thanks for the suggestion. We will further revise this section.

---

> ### Author Response · Authors · 2024-08-09
> **About our responses**
>
> Dear Reviewers,
>
> Thanks for your efforts and the constructuve suggestions for this paper. We have provided the response based on your comments.
>
> Please read our responses and feel free to discuss with us during the reviewer-author discussion stage, if you have any concerns for our response.
>
> Best Regards,
>
> The authors

---

> > ### Comment · Reviewer_94wD · 2024-08-12
> >
> > The responses address my concerns about this paper. Thus, I keep to recommend the acceptance and improve the score of this paper.

---

### Official Review · Reviewer_DgD7 · 2024-07-13

**Soundness:** 3
**Presentation:** 3
**Contribution:** 3
**Rating:** 7
**Confidence:** 4

**Summary:**

The authors propose a new GNN-based representation learning method (HC-GAE), that can abstract effective local node features and global graph features. These features can be used for both node and graph classification. The new HC-GAE method consists of two main computational modules, they are the encoder associated with the hard node assignment, and the decoder associated with the soft assignment. Moreover, the deep representation in the middle layer can be seen as the global graph features, and the output in the last layer can be seen as the local node features. All these features encapsulate bidirectionally hierarchical structural features of the original sample graph based on the hierarchical strategy. Finally, the authors also propose a new loss function for integrating the information from either the encoder or the decoder.

**Strengths:**

1. The idea of this work seems novel and interesting for me, performing the local convolution operation within the separated subgraphs through the hard node assignment not only addresses the over-smoothing problem, but also forms hierarchical representation for the graphs.
2. The descriptions are clear, the experiments demonstrate the effectiveness, and the new model is technical sound.

**Weaknesses:**

Overall, the writing is easy to understand, but I see minor typos or grammar mistakes in Section 3 and 4. I didn’t check these in details, the authors should carefully correct them for the final manuscript.

I feel a little strange for the loss function. Because the function tends to minimize the differences of either the structures or the node features between the input (for encoder) and the output (decoder). As I see, if the authors use the output for the node classification, why you do not directly use the original input node features? Or, foe the node classification you eliminate the effects from the node features for the loss function. This is not very clear.

Appendix B, what do you mean for Ghazan? I didn’t see any explanation of this word. Is it a kind of alternative method or others?

The authors discuss some theoretical advantages of the proposed method in Sec 3.4. But they don’t discuss any reason about why the proposed method performs well in Sec 4. They just simply show the accuracies are better. Some more analysis is needed.

**Questions:**

See the questions I asked in weakness.

**Limitations:**

Yes

---

> ### Author Rebuttal · Authors · 2024-08-07
>
> Q1: I feel a little strange for the loss function. Because the function tends to minimize the differences of either the structures or the node features between the input (for encoder) and the output (decoder). As I see, if the authors use the output for the node classification, why you do not directly use the original input node features? Or, foe the node classification you eliminate the effects from the node features for the loss function. This is not very clear.
>
> A1: We are sorry that we didn’t explain the loss function for more details. For graph classification, the proposed model needs to simultaneously reconstruct the topological structures (i.e., the adjacency matrices) and the node feature information (i.e., the node feature matrices). On the other hand, for the node classification, we only need to reconstruct the structure information. We will explain these more clear in the manuscript.
>
> Q2: Appendix B, what do you mean for Ghazan? I didn’t see any explanation of this word. Is it a kind of alternative method or others?
>
> A2: We are sorry for the typos, in fact we should use the HC-GAE. We will correct this problem.
>
>
> Q3: The authors discuss some theoretical advantages of the proposed method in Sec 3.4. But they don’t discuss any reason about why the proposed method performs well in Sec 4. They just simply show the accuracies are better. Some more analysis is needed.
>
> A3: Thanks for the constructive suggestion. The theoretical reasons for the effectiveness are twofold. First,  the proposed HC-GAE is based on the new subgraph convolution framework. Specifically, we propose to employ the hard node assignment to assign the nodes into separated clusters, resulting in separated subgraphs. The convolution operation is performed within each individual subgraphs, and the node information cannot be propagated between different subgraphs, this can thus significantly reduce the over-smoothing problem and improve the performance. Second, the proposed HC-GAE can simultaneously reconstruct both the topological structures and the node features through the hierarchical strategy associated with the node assignment operation. Moreover, it will not drop any node information, reducing the topological information loss, effectively extracting bidirectionally hierarchical structural features of the original sample graph. We will add these theoretical discussion in the manuscript associated with the classification accuracies, following the the reviewer’s suggestion.

---

> > ### Comment · Reviewer_DgD7 · 2024-08-12
> >
> > The authors’ responses seem reasonable for my questions. Please revise the paper as promised in the rebuttal.

---

> ### Author Response · Authors · 2024-08-09
> **About our responses**
>
> Dear Reviewers,
>
> Thanks for your efforts and the constructuve suggestions for this paper. We have provided the response based on your comments.
>
> Please read our responses and feel free to discuss with us during the reviewer-author discussion stage, if you have any concerns for our response.
>
> Best Regards,
>
> The authors

---

### Decision · Program_Chairs · 2024-09-25

**Decision:**

Accept (poster)

**Comment:**

This paper appears solid enough to be above the NeurIPS bar. The paper targets a crucial problem in GNNs (over-smoothing), proposing a sufficiently innovative, strongly performant, and flexible (both in node and graph classification) solution.

Overall, this looks like a valid contribution to the field; I recommend acceptance, although the authors are strongly advised to implement the suggestions of the reviewers in the final version.

**Note to SAC** I will add a summary of the paper once the final decision has been taken.